# Circulating HHIP Levels in Women with Insulin Resistance and PCOS: Effects of Physical Activity, Cold Stimulation and Anti-Diabetic Drug Therapy

**DOI:** 10.3390/jcm12030888

**Published:** 2023-01-22

**Authors:** Xin Zhou, Yanping Wang, Wenyun Chen, Hongmin Zhang, Yirui He, Han Dai, Wenjing Hu, Ke Li, Lili Zhang, Chen Chen, Gangyi Yang, Ling Li

**Affiliations:** 1The Key Laboratory of Laboratory Medical Diagnostics in the Ministry of Education and Department of Clinical Biochemistry, College of Laboratory Medicine, Chongqing Medical University, Chongqing 400042, China; 2Department of Endocrinology, The Second Affiliated Hospital, Chongqing Medical University, Chongqing 400010, China; 3Department of Endocrinology, The First People’s Hospital of Chongqing Liang Jiang New Area, Chongqing 401121, China; 4Endocrinology and Metabolism, SBMS, The University of Queensland, Brisbane 4072, Australia

**Keywords:** serum human hedgehog-interacting protein (HHIP), insulin resistance (IR), polycystic ovary syndrome (PCOS)

## Abstract

Serum human hedgehog-interacting protein (HHIP) concentration is associated with diabetes. However, the relationship between HHIP and polycystic ovary syndrome (PCOS) or abnormal sex hormones remains unknown. This study was an observational cross-sectional study, with additional short-term intervention studies and follow-up studies. Bioinformatics analysis was performed to explore the association of PCOS with metabolic-related genes and signaling pathways. OGTT and EHC were performed on all participants. Lipid infusion, cold exposure, and 45-min treadmill test were performed on all healthy women. A total of 137 women with PCOS were treated with metformin, GLP-1RA, or TZDs for 24 weeks. Serum HHIP levels were higher in insulin resistance (IR) and PCOS women. Circulating HHIP levels were significantly correlated with adiponectin (Adipoq) levels, obesity, IR, and metabolic indicators. A correlation presented between HHIP and DHEA-S, FAI, SHBG, and FSH. Serum HHIP levels were significantly elevated by oral glucose challenge in healthy women, but not affected by EHC. Lipid infusion decreased serum HHIP levels, while cold exposure increased HHIP levels in healthy women. GLP-1RA and TZD treatment reduced serum HHIP levels in PCOS women, while metformin treatment did not affect HHIP levels. HHIP may be a useful biomarker and novel drug target for PCOS and IR individuals.

## 1. Introduction

Polycystic ovary syndrome (PCOS) is a common disease in young women that is mainly characterized by reproductive and endocrine metabolic disorders. In recent decades, the incidence rate of PCOS has been increasing on a yearly basis and 2.4–11.9% of the cases occur in women of reproductive age [1,2]. In addition to hyperandrogenemia and reproductive disorders, PCOS is often accompanied by obesity, insulin resistance (IR)/hyperinsulinemia and metabolic disorders [3,4]. Obesity and IR further exacerbates reproductive and metabolic abnormalities [5,6]. At present, the exact pathogenesis of PCOS remains unknown, and there are no standard diagnostic criteria. Therefore, identifying new biomarkers is of great clinical significance for accurate diagnosis and treatment of PCOS.

Human hedgehog-interacting protein (HHIP) is an important factor in the maintenance of regeneration of various tissues in embryogenesis [7]. It is also pivotal for the endocrine and exocrine function of the pancreas [8,9,10,11]. HHIP is a trans-membrane glycoprotein containing 700 amino acid residues that negatively regulate the hedgehog pathway [12]. HHIP was originally identified as an inhibitor of the HH ligand, which can regulate cell functions, such as pathologic angiogenesis and muscle development, through classical or non-classical HH pathways [13,14,15,16,17,18,19,20,21]. It has been found that HHIP can inhibit insulin secretion in high-fat diet (HFD)-fed mice by promoting islet ß cell dysfunction [22]. In addition, an analysis on a diabetes-related genome-wide database found that HHIP mRNA expression in the islets of mice with obesity (ob/ob mice) was significantly increased compared with lean mice [23]. Further, recently, a study found that the serum HHIP concentration increased significantly in patients with impaired fasting glucose (IFG), impaired glucose tolerance (IGT) and newly diagnosed type 2 diabetes mellitus (T2DM). Furthermore, HbA1c and fasting blood glucose (FBG) levels were independently correlated with circulating HHIP concentration [24]. Therefore, serum HHIP concentration is considered to be associated with diabetes and metabolic abnormalities.

It is well known that 70–80% of women with PCOS have IR and metabolic disorders [2,25]. Thus, it is important to investigate the relationship between HHIP and the occurrence and development of PCOS. In the current study, we hypothesized that HHIP is associated with PCOS and metabolic abnormalities. To validate this hypothesis, we measured circulating HHIP levels in women with PCOS and analyzed its relationship with other metabolic indicators.

## 2. Materials and Methods

### 2.1. Bioinformatic Analysis

The gene expression database (GEO, https://www.ncbi.nlm.nih.gov/geo/, accessed on 12 August 2019) was searched by the keyword “Polycystic Ovary Syndrome”, selecting “Homo sapiens” in top organizations, selecting “study type” as “expression profiling by array”, and selecting “entry type” as “series” to obtain 28 results. By screening the title, introduction, and platform information, we focused on (1) the expression profile data in PCOS and control group; (2) the adipocyte or adipose stem cell (ASC) data related to metabolism; and (3) the secretory protein genes in top 20 differentially expressed genes (DEGs). Finally, the GEO dataset (GSE 124226) was obtained.

### 2.2. Study Population and Anthropological Examination

The current study screened 550 women and finally included 469 individuals, including 195 women with PCOS, 117 women with IR, and 157 healthy individuals. The diagnostic criteria of IR were M-value > 6.286 during clamping [26]. The diagnosis of PCOS was based on the 2003 Rotterdam consensus standard, as previously reported [27]. PCOS and IR were newly diagnosed in these women. All women had no other diseases and were taking no drugs before this study. In the first 6 months of the study, the subjects did not take drugs that might interfere with the study, such as oral contraceptives, insulin sensitizers, antiandrogens, or glucocorticoids. Individuals with hyperandrogenemia caused by other causes than PCOS, such as hyperprolactinemia, hypothyroidism, Cushing’s syndrome, and congenital adrenal hyperplasia, were excluded from this study. Age-matched healthy women without clinical evidence of major diseases were recruited from an unselected population that underwent a routine medical check-up and were used as the controls. Healthy control individuals did not use any drugs. All participating women gave informed written consent before the study. The study was approved by the Human Ethics Committee of Chongqing Medical University. This study was registered as ChiCTR2000032494 and ChiCTR-IIR-16007901. The anthropological examination was performed at 8:00 a.m. including body weight (BW), body mass index (BMI), blood pressure, waist circumference (WC), etc., following a standard procedure.

### 2.3. Oral Glucose Tolerance Test (OGTT), Biochemical Parameters, and Sex Hormones Determination

After 12 h of fasting, participating women were given 75 g of glucose orally. OGTT was performed as previously reported [28]. During the OGTT, blood samples were taken at the designated time to determine blood glucose, biochemical indexes, and HHIP. Blood glucose, HbA1c, insulin, blood lipid, free fatty acid (FFA), and sex hormone measurements were measured as described previously [29].

### 2.4. Euglycemic-Hyperinsulinemic Clamp Experiment (EHC)

To assess glucose metabolism and insulin sensitivity in vivo, EHC was performed in all individuals, as previously reported [29]. Insulin (1 mU/kg/min, Novo Nordisk, Copenhagen, Denmark) was continuously infused into circulation for 2 h, and the blood glucose was adjusted and clamped to about 5.5 mmol by 20% glucose infusion with a changing rate. Insulin sensitivity was determined by the glucose infusion rate (GIR) during the last 45 min of the EHC. The glucose disposal rate (GDR) was defined as the GIR during the stable state of the EHC and was related to body weight (M-value). Blood samples were collected at the specified time points (0, 80, 100, and 120 min) for measuring HHIP and other indicators.

### 2.5. Lipid Infusion Combined with EHCs

To investigate the effect of FFA-induced IR on circulating HHIP levels, we engaged in lipid infusion combined with EHC. Twenty-five healthy subjects (11 men and 14 women, age 26.7 ± 1.9 years; BMI, 22.0 ± 1.1 kg/m^2^) were enrolled to take part in the test. After 10 h of fasting, the individual received a 240 min infusion of lipid (20% Intralipid, Pharmacia and Upjohn)/heparin solution. 120 min after the start of lipid infusion, an EHC began until the end of the experiment. The rate of lipid infusion was 1.5 mL/min. Blood samples were obtained at indicated time points.

### 2.6. Exercise Intervention Study

A total of 20 healthy subjects, 11 men and 9 women, were included in the exercise intervention study. After fasting for 12 h, at 8:00 AM, the individual engaged in a treadmill experiment for 45 min, followed by a 120-min rest. Blood was drawn at the indicated time to determine HHIP concentration, as previously reported [30].

### 2.7. Cold-Exposure Procedure

To investigate the effect of cold-induced adaptive thermogenesis on circulation HHIP levels, 20 healthy individuals (11 males and 9 females) participated in the cold-exposure project. After a night of fasting, participants lay in bed with light clothes under a water-circulating cooling blanket. (ThermoBlanket, P&C-AII, Hengbang Technology, China). To avoid shaking, electromyography (EMG, je-tb0801, China Ellie Technology) is used to adjust the temperature. Blood samples were taken after exposure to 27 ℃ for 30 min. The water temperature drops to 12 ℃ every 2 min. The temperature was maintained at 12 ℃ for 5 min and then backed to 27 ℃ again. Blood samples were taken at the indicated times [31].

### 2.8. GLP-1RA, Metformin and Thiazolidinedione Intervention Study

154 newly diagnosed women with PCOS were randomly screened for the drug study. Finally, 137 individuals were enrolled in the study (age 26.9 ± 4.9 years; BMI 26.0 ± 4.3 kg/m^2^), including 52 women treated by glucagon-like peptide-1 receptor agonists (GLP-1RA, liraglutide) group, 45 women treated by metformin group and 40 women treated by thiazolidinedione (TZD) for 24 weeks as three groups. All participants were given written informed consent for the side effects of metformin, TZD and liraglutide (lira) at the beginning of the drug intervention study. Metformin was started at 0.5 g twice daily and gradually increased to 1.5 g twice daily. Lira was increased from 0.6 mg to 1.8 mg once daily. Rosiglitazone (TZD) started at 4 mg and was titrated to 8 mg once daily. All women were asked to keep their former lifestyle and dietary habits during the study. Before and 24 weeks after treatment, anthropometric and biochemical parameters, sex hormones, HHIP and adipoq levels were measured to assess the efficacy and the effect of the treatment.

### 2.9. HHIP and Adiponectin (Adipoq) Measurements

Serum samples were assayed for HHIP (Meimian Biotechnology, Yancheng, Jiangsu, China) and adipoq (sk00010-01, Aviscerabio science Inc., USA) using the ELISA kits. Kits for HHIP had intra- and inter-assay coefficients of variation (CV) less than 10% and 8%, respectively. The minimum detection level for human HHIP is 0.1 ng/mL, with a detection range of 0.625–20 ng/mL. The method had high sensitivity and good specificity for human HHIP without obvious cross-reactivity or interference. Intra- and inter-assay CV for adipoq were < 9% and < 11%, respectively. Serum Adipoq was determined as previously reported [31].

### 2.10. Calculation and Statistical Analysis

Homeostasis model assessment (HOMA)-IR was calculated as follows: HOMA-IR=fasting insulin (FIns, Mu/L) × FBG mmol/L)/22.5 [28]. The glucose (AUC_glucose_) and insulin (AUC_insulin_) areas under the curve were calculated using the trapezoidal rule. The free androgen index (FAI) was calculated as FAI=[testosterone (TEST)/sex-hormone binding globulin (SHBG)] ×100 [32]. Body adiposity index (BAI) was calculated as $\mathrm{BAI}=\mathrm{hip}\;\mathrm{circumference}/{\mathrm{height}}^{1.5}\;-\;18$ [33]. Visceral adiposity index (VAI)=waist circumference (WC)/(36.58+1.89 ×BMI)
× tri-glyceride (TG)/0.81 × 1.52/high-density lipoprotein cholesterol (HDL-C) [34]. The statistical analysis was performed on SPSS version 22.0 (SPSS, Armonk, NY, USA). Kolmogorov–Smirnov test was used for the distribution of data. The *t-test* (normal distribution data) or Mann–Whitney *U* test (abnormal distribution data) was used for comparison between groups. Spearman correlation analyses were performed to calculate the correlation between HHIP levels and other parameters. The binary logistic regression analysis was used to control the possible confounding variables and to assess the relationship between HHIP and PCOS and IR. The receiver operating curves (ROCs) were drawn with sensitivity and (100 specificities) as the vertical axis and horizontal axis, respectively. The area under the ROC curve (AUC) of 95% confidence interval (CI) was calculated to estimate the diagnostic ability of HHIP for PCOS and IR. Data were shown as means ± standard deviation (SD), or median (interquartile range). *p* < 0.05 was considered significant compared with the controls.

## 3. Results

### 3.1. Bioinformatics Data Analysis

To search for genes related to PCOS, we obtained the gene expression profiles of abdominal adipose stem cells (ASCs) from four PCOS women and four age- and BMI-matched healthy women from the Gene Expression Omnibus (GEO) dataset (GSE124226). We screened differentially expressed genes (DEGs) with a change of expression level ≥ 2 times by using limma software package (*p* value < 0.05; log Fold Change = 1) (Figure 1a). The heatmap showed 79 DEGs from GSE124226 (Figure 1b). According to |Log2 FC|, we found the top 20 DEGs. Among them, HHIP expression changed most obviously (Figure 1c). Therefore, bioinformatics analysis showed that HHIP was related to PCOS.

### 3.2. Clinical, Hormonal and Metabolic Parameters in the Study Population

Clinical, hormonal and metabolic parameters in the study population were shown in Table 1. In IR and PCOS women, BMI, waist-to-hip ratio (WHR), systolic blood pressure (SBP), total cholesterol (TC), low-density lipoprotein cholesterol (LDL-C), FBG, 2h-blood glucose after glucose overload (2h-BG), FIns, 2h-insulin after glucose overload (2h-Ins), HbA1c, AUC_glucose_, AUC_insulin_, HOMA-IR, and BAI were increased significantly compared with healthy controls, while M-value was lower, suggesting metabolic disorders and obesity in IR and PCOS groups. Compared with the IR group, PCOS women had higher TG levels, AUC_insulin_ and M-values, and lower diastolic blood pressure (DBP), TC, LDL-C and FFA levels (Table 1). In IR and PCOS women, dehydroepiandrosterone sulfate (DHEA-S) and FAI were significantly higher than those in healthy women, while follicle-stimulating hormone (FSH) and SHBG were significantly lower (Table 1). Furthermore, in PCOS women, luteinizing hormone (LH), TEST and FAI were significantly higher than those in the IR group. Also, the IR group had lower TEST levels than other groups (Table 1).

### 3.3. Circulating HHIP and Adipoq Levels and the Association of HHIP with Other Indicators in the Study Population

In this cross-sectional study, we simultaneously measured adipoq and HHIP levels in IR, PCOS, and healthy women. The distribution of circulating HHIP levels in healthy controls ranged from 4.25 to 16.7 μg/L, and 90% of the healthy population was between 4.9 to 14.5 μg/L (Figure 2a,b). We found that HHIP levels increased slightly with the increase of age (Figure 2c). In addition, our results showed that the levels of circulating HHIP in IR and PCOS women were significantly higher than those in healthy controls, while serum adipoq levels decreased significantly (Table 1 and Figure 2d,e). Our linear correlation analysis showed that HHIP was significantly positively correlated with BMI (*r* = 0.39, *p* < 0.01), WHR (*r* = 0.35, *p* < 0.01), SBP (*r* = 0.27, *p* < 0.01), FBG (*r* = 0.40, *p* < 0.01), 2h-BG (*r* = 0.43, *p* < 0.01), FIns (*r* = 0.41, *p* < 0.01), 2h-Ins (*r* = 0.42, *p* <0.01), TG (*r* = 0.10, *p* < 0.05), TC (*r* = 0.22, *p* < 0.01), LDL-C (*r* = 0.19, *p* < 0.01), HbA1c (*r* = 0.28, *p* < 0.01), HOMA-IR (*r* = 0.43, *p* < 0.01), AUC_glucose_ (*r* = 0.43, *p* < 0.01), AUC_insulin_ (*r* = 0.38, *p* < 0.01), VAI (*r* = 0.16, *p* < 0.01), BAI (*r* = 0.33, *p* < 0.01), DHEA-S (*r* = 0.17, *p* < 0.01), and FAI (*r* = 0.17, *p* < 0.01), while negatively correlated with SHBG (*r* = −0.22, *p* < 0.01), FSH (*r* = −0.18, *p* < 0.01), and HDL-C (*r* = −0.14, *p* < 0.01) (Figure 2f). Moreover, circulating HHIP levels were significantly negatively correlated with adipoq (*r* = −0.47, *p* < 0.01) and M-value (*r* = −0.45, *p* < 0.01) (Figure 2f), further suggesting that HHIP may be a biomarker related to IR. Multiple stepwise regression analysis showed that M-value, adipoq, FBG and luteinizing hormone (LH) were independent influencing factors (Figure 2g), and the multiple regression equation is $Y_{\log\;\mathrm{HHIP}}=1.086-0.012X_{M\text{-}\mathrm{value}}-0.002X_{\mathrm{Adipoq}}+0.029X_{\mathrm{FBG}}+0.002X_{\mathrm{LH}}$. In addition, multiple logistic regression analysis showed that circulating HHIP was significantly associated with PCOS and IR even if age and sex were controlled (Table 2). Using the Row Mean Score and Cochran Armitage test, we found that serum HHIP concentration was independently correlated with PCOS and IR and showed a linear increasing trend, and HHIP was independently associated with PCOS and IR (Appendix A).

Results of multivariate logistic regression analysis are presented as the odds ratio of being in IR and PCOS status increase in HHIP. CI, confidence interval; OR, odds ratio. PCOS, polycystic ovary syndrome; IR, Insulin resistance; BMI, body mass index; WHR, waist-to-hip ratio; SBP, systolic blood pressure; DBP, diastolic blood pressure; TG, triglyceride; TC, total cholesterol; HDL, high-density lipoprotein cholesterol; LDL, low-density lipoprotein cholesterol; FFA, free fatty acids.

Next, we divided HHIP and adipoq concentrations in PCOS subjects and normal controls into four quartiles (quartile 1, <9.4 μg/L; quartile 2, 9.4–12.0 μg/L; quartile 3, 12.0–15.6 μg/L and quartile 4, >15.6 μg/L for HHIP, and quartile 1, <23.1 mg/L; quartile 2, 23.1–33.9 mg/L; quartile 3, 33.9–45.3 mg/L; quartile 4, >45.3 mg/L for adipoq). We found that when the individual HHIP concentration was in quartiles 2, 3, and 4, the risk of PCOS was higher than that in quartile 1 (95% CI 2.079–4.809; 95% CI 2.875–6.726; 95% CI 8.521–33.251, *p* < 0.01) (Figure 2h). Individuals with adipoq concentrations in quartile 2, 3, and 4 had a higher risk of PCOS than those in the quartile 1(95% CI 0.568–1.330; 95% CI 0.200–0.429; 95% CI 0.091–0.223, *p* < 0.01) (Figure 2h). In addition, we divided the IR and healthy populations into four quartiles according to HHIP and adipoq concentrations (quartile 1, <8.7 μg/L; quartile 2, 8.7–11.7 μg/L; quartile 3, 11.7–15.1 μg/L; quartile 4, >15.1μg/L for HHIP and quartile 1, <23.2 mg/L; quartile 2, 23.2–37.4 mg/L; quartile 3, 37.4–48.1 mg/L; quartile 4, < 48.1 mg/L for adipoq) (Figure 2i). When individual HHIP levels were 2, 3, and 4, the risk of IR was significantly higher than those of quartile 1. Adipoq quartiles were 2, 3 and 4, and the risk of IR was significantly lower than those of quartile 1 (Figure 2i). We further performed a ROC curve analysis to evaluate the prediction of HHIP for PCOS and IR occurrence. ROC analysis showed that AUC for PCOS was 0.865, sensitivity was 74.4%, specificity was 79.0%, cut-off point value was 11.9 μg/L (Figure 2j) and AUC for IR was 0.904, the sensitivity was 71.8%, the specificity was 91.7%, and the cut-off value was 13.4μg/L (Figure 2k). Our results showed that HHIP and adipoq can both predict PCOS and IR (Figure 2j,k).

### 3.4. Alteration of Circulating HHIP Levels during OGTT and EHC Experiments

To observe the effects of blood glucose and insulin levels on circulating HHIP, we first performed the OGTT experiment (Figure 3a). During the OGTT, for oral glucose challenge, both circulating HHIP (from 9.8 ± 3.2 to 11.5 ± 4.1 μg/L for 30 min, and then to 9.8 ± 3.7 μg/L for 60 min) and adipoq (from 42.9 ± 14.2 to 54.3 ± 11.5 for 30 min, and then to 36.8 ± 15.1 mg/L for 60 min) levels were significantly elevated at 30 min in healthy controls (Figure 3b). In IR and PCOS women, the changes of HHIP and adipoq were similar to those in healthy individuals (Figure 3c, d). In addition, AUC_adipoq_ in women with IR and PCOS was significantly lower, while AUC_HHIP_ was significantly higher (Figure 3e).

Next, we undertook an EHC study in healthy controls, IR and PCOS women. The EHC experimental process is shown in Figure 4a. During the EHC, blood glucose was stabilized at about 5.5 mM, and insulin level gradually increased. During the steady state of EHC, serum adipoq levels in the control and PCOS group were significantly decreased (from 48.9 ± 14.8 to 22.1 ± 9.3 mg/L for control group; from 25.9 ± 8.5 to 18.2 ± 5.2 mg/L for PCOS group), while no change in IR group (Figure 4b,c), while HHIP increased slightly in the PCOS group (Figure 4b,c).

### 3.5. Effect of Lipid-Induced IR, Physical Activity and Cold-Exposure Test on Circulating HHIP Levels In Vivo

To observe the effect of increased FFA concentration on serum HHIP level, we performed the lipid infusion and EHC, physical activity and cold-exposure test in healthy individuals (Figure 5a, b upper panel). We found that increased FFA levels led to a decrease in HHIP levels during lipid infusion (from 12.3 ± 3.4 to 11.2 ± 4.0 μg/L at 80 min), suggesting that acute-IR may inhibit HHIP release (Figure 5b below panel). However, the 45 min treadmill test did not result in changes in serum HHIP levels (Figure 5c). In addition, to exclude the effect of gender on circulating HHIP levels in cold-induced adaptive thermogenesis, we performed a cold-stimulation test in both men and women. We found that cold-stimulation led to an increase in circulating HHIP levels (from 11.9 ± 2.4 to 13.0 ± 3.4 μg/L vs. pre-stimulation) (Figure 5d), suggesting that HHIP secretion and release may be affected by brown adipose tissue (BAT) thermogenesis.

### 3.6. Effects of Antidiabetic Drug on Circulating HHIP Levels and Other Indicators Pre- and Post-Treatment

To investigate the effect of antidiabetic drugs on serum HHIP level and other indicators, we used three antidiabetic drugs to treat PCOS patients (Figure 6a). The changes of serum HHIP levels and other indicators in PCOS women before and after anti-diabetic drug treatment were shown in the Appendix A. We found that after treatment with three drugs, WHR, TC, LDL-C, FFA, FBG, 2h-BG, FIns, 2h-Ins, HbA1c, AUC_glucose_, AUC_insulin_ and HOMA-IR in PCOS women decreased significantly, while M-value increased significantly. In addition, GLP-1RA treatment significantly reduced BMI, TG and SBP levels, and metformin slightly reduced BMI (Appendix A). FAI was significantly decreased after the three drug treatment. Furthermore, metformin treatment resulted in a significant decrease in DHEA-S and a significant increase in SHBG. GLP-1RA treatment resulted in an increase in estradiol (E2) levels and a significant decrease in LH, Prog and TEST. TZD treatment increased E2 and SHBG levels, but decreased TEST levels (Appendix A). Importantly, the circulating levels of adipoq in the three groups were significantly increased after treatment (Appendix A), but there was no significant difference among the three groups (Figure 6b above panel). GLP-1RA and TZD treatment reduced serum HHIP levels, while metformin treatment had no effect (Appendix A and Figure 6c–e).

## 4. Discussion

Recently, several human and animal studies have found that HHIP is associated with metabolic diseases such as T2DM and obesity [22,23,24]. However, the relationship between HHIP and PCOS has not been reported. To the best of our knowledge, this study is the first to report that HHIP is associated with metabolic disorders and PCOS through bioinformatics analysis. Our results showed that circulating HHIP levels were significantly elevated in women with IR and PCOS, whereas serum adipoq levels were significantly decreased. Further, in the current cohort, circulating HHIP levels were significantly associated with glucose and lipid metabolism disorders, obesity, IR and PCOS, and were significantly negatively correlated with serum adipoq levels and the M-value of EHC. During OGTT, the serum HHIP concentration increased significantly with elevated blood glucose levels in the normal population. According to the results of EHC, hyperinsulinemia resulted in a significant decrease in serum adipoq levels, while circulating HHIP levels did not change in normal controls. Further, lipid-induced IR led to a slight decrease in serum HHIP levels in healthy controls, while cold stimulation led to an increase in circulating HHIP levels. Our results also demonstrate that treatment with anti-diabetic drugs significantly improved IR and reduced FAI levels in women with PCOS, and GLP-1RA and TZD treatment reduced serum HHIP levels. Collectively, these findings indicate that HHIP may be a useful biomarker in individuals with IR and PCOS.

Recently, it has been reported that HHIP mRNA expression in the islets of mice with obesity (ob/ob mice) was significantly increased [23], and that the serum HHIP concentrations were significantly increased in patients with IFG, IGT, and newly diagnosed T2DM [24]. Consistent with previous results in humans and animals, in our cross-sectional study, we found that the serum HHIP concentration was significantly increased in women with PCOS and IR. In addition, the levels of circulating HHIP were significantly related to obesity, IR indicators, and glucose and lipid metabolism. These findings indicate that obesity, IR, and metabolic disorders are strong predictors of serum HHIP concentrations. Further, we found that serum HHIP levels were elevated in individuals with IR and PCOS, and these findings indicate that HHIP may be a biomarker of metabolic disorders and a potential predictor of PCOS. Given that the patterns of serum HHIP levels and their relationship with obesity, IR, and metabolic parameters were in contrast to those of adipoq in the women with IR and PCOS included in our cohort, we hypothesized that the role of HHIP in metabolic disorders may be different from that of adipoq, that is, HHIP promotes rather than improves IR.

In the intervention study, physical activity in healthy populations did not lead to changes in serum HHIP levels; this suggests that the muscle may not be the main organ involved in HHIP secretion. Interestingly, cold exposure resulted in a significant increase in circulating HHIP levels in healthy women. It has been reported that cold exposure stimulates the sympathetic nervous system (SNS) to promote BAT activation, which in turn affects glucose and lipid metabolism [32]. In fact, previous studies have found that under conditions of physiological stress, such as cold exposure, the in vivo cytokine levels (IL-1ß) increase [33]. The reason for the increase in cytokine levels in response to stress is not clear, but it may be caused by an increase in the levels of stress hormones such as adrenaline. This line of research needs further study.

Anti-diabetic drugs, including GLP-1RA, TZDs and metformin, can reduce serum insulin levels and improve IR. These drugs are widely used in PCOS patients to improve metabolic and endocrine disorders [34]. Recently, our research and other studies have found that anti-diabetic drug treatment can increase the circulating levels of certain cytokines, such as secreted frizzled-related protein-5 (Sfrp5) and omentin-1 [35,36].

In the current study, we found that metabolic disorders and IR in patients with PCOS were significantly improved after treatment with GLP-1RA, TZDs, and metformin. Among the three drugs, GLP-1RA resulted in the most prominent weight loss, and it was followed by metformin. Further, treatment with all three drugs significantly reduced FAI, but only GLP-1 and TZDs treatments reduced TEST levels. This implies that GLP-1 and TZDs may have a better effect in terms of improving hyperandrogenemia, menstrual cyclicity, and ovulation in women with PCOS.

Changes in steroid metabolism are known to occur in women with PCOS. Therefore, the findings of this study are not completely consistent with previously published studies. As we did not observe a significant decrease in TEST levels after metformin treatment, this implies that metformin treatment has no obvious effect on steroid metabolism. The reason for the difference in the findings of our study and previous studies remains unknown, and it may be related to the sample size, experimental conditions, and differences between the enrolled populations.

With the improvement of metabolic disorder and hyperandrogenemia, the level of circulating adipoq was found to increase significantly in all three treatment groups, but a significant decrease in serum HHIP levels was observed only in the TZDs and GLP-1RA groups. These results suggest that while all three drugs can promote adipoq secretion, GLP-1RA and TZDs can also inhibit HHIP secretion. This may imply that GLP-1RA and TZDs have a stronger effect on improving insulin sensitivity than metformin. The mechanism via which treatment with GLP-1RA and TZDs led to the increase in serum HHIP levels is unknown. However, the effect of GLP-1RA and TZDs on serum HHIP levels may be achieved via regulation of other circulating factors such as insulin and androgen. Further research is needed to shed light on the associated mechanisms.

Our study has some limitations that need to be mentioned: (1) This is a cross-sectional study that does not reflect the long-term course of changes in circulating HHIP levels in the study population. Therefore, further follow-up studies are necessary. (2) The participants were mainly from the Han Chinese population, so our results may not apply to other populations. (3) Our pharmacological intervention included a self-controlled study before and after treatment. Therefore, further randomized, double-blind, placebo-controlled trial studies are needed to compare the efficacy of GLP-1, metformin, and TZDs and confirm the present findings.

## 5. Conclusions

This cross-sectional cohort study showed that women with PCOS and IR women have higher than normal HHIP levels and lower than normal adipoq levels. Further, circulating HHIP levels were found to be associated with glucose and lipid metabolism and insulin sensitivity, and were not affected by 45-min aerobic exercise but increased after cold exposure in healthy women. Finally, treatment with GLP-1RA and TZDs in women with PCOS improved IR, increased serum adipoq levels, and decreased HHIP levels. Thus, our data indicate that HHIP might be a potential biomarker for the identification of high-risk candidates among women with IR and PCOS.

## Figures and Tables

**Figure 1 jcm-12-00888-f001:**
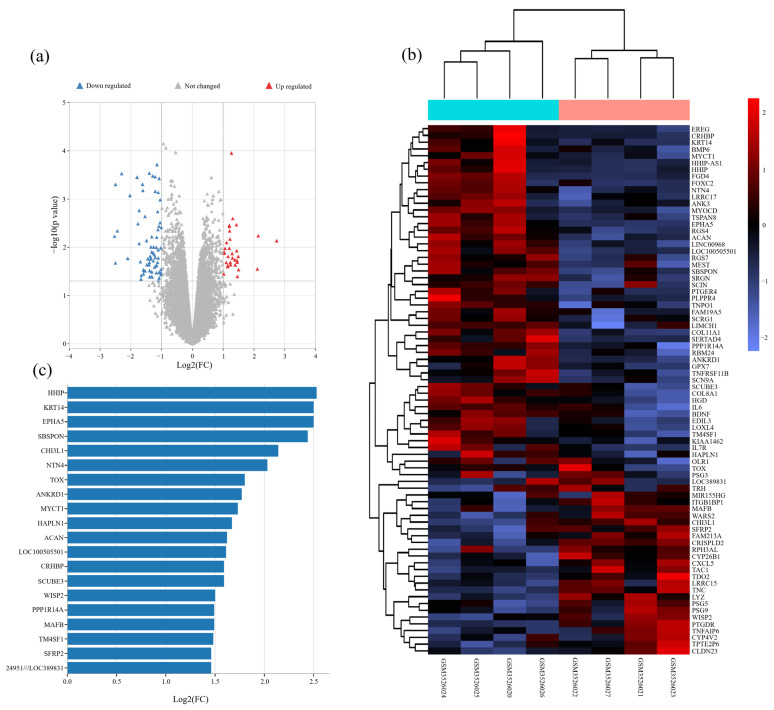
Bioinformatics analysis for PCOS related genes and signaling pathways in PCOS women. (**a**) Volcano map of DEGs from abdominal adipose stem cells of PCOS women and healthy women. (**b**) Clustering analysis of genes from abdominal adipose stem cells of PCOS women and healthy women. (**c**) The sequence of the top 20 genes with the highest degrees.

**Figure 2 jcm-12-00888-f002:**
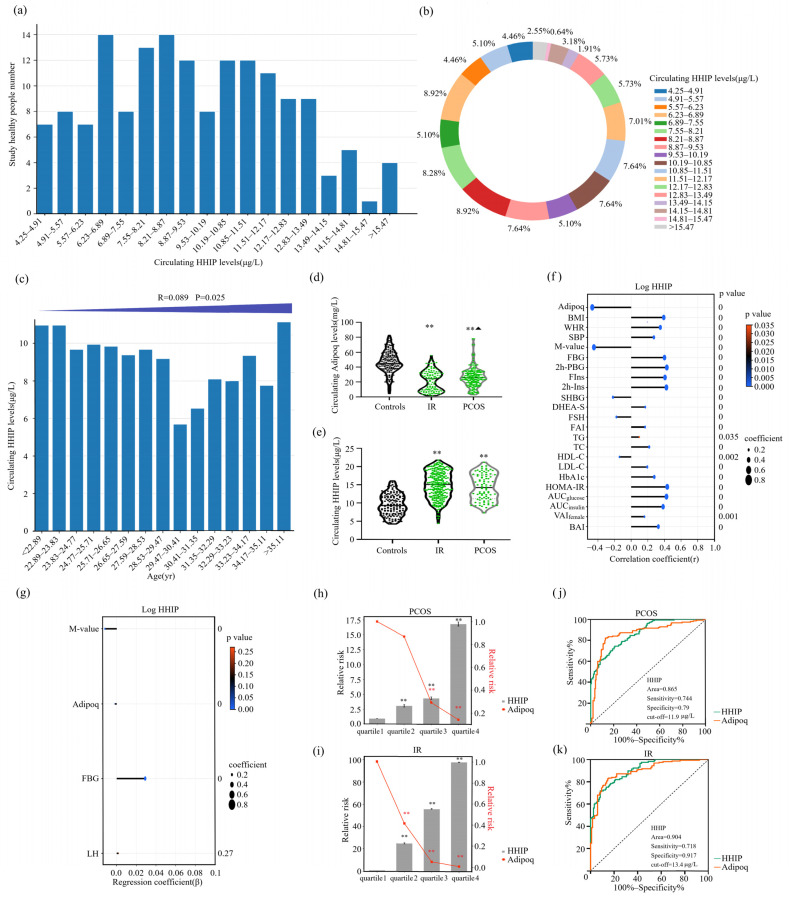
Serum HHIP level in the study population and its relationship with other parameters. (**a**) Distribution of serum HHIP in healthy women. (**b**) Percentage of serum HHIP concentration distribution in normal women. (**c**) Distribution of serum HHIP concentration in normal women of different ages. (**d**,**e**) Serum HHIP and adipoq concentration in IR, PCOS and healthy women. (**f**) Linear correlation analysis for serum HHIP and other variables. (**g**) Multiple regression analyses for serum HHIP and other variables. (**h**) Prevalence of elevated PCOS in different quartiles of HHIP and adipoq. (**i**) Prevalence of elevated IR in different quartiles of HHIP and adipoq. (**j**,**k**) ROC curve analyses for the prediction of PCOS (**j**) and IR (**k**) according to serum HHIP levels. Data are expressed as mean ± SD or median (Interquartile Range). ** *p* < 0.01 vs. controls or. quartile 1; ^▲^
*p* < 0.05 compared with IR group.

**Figure 3 jcm-12-00888-f003:**
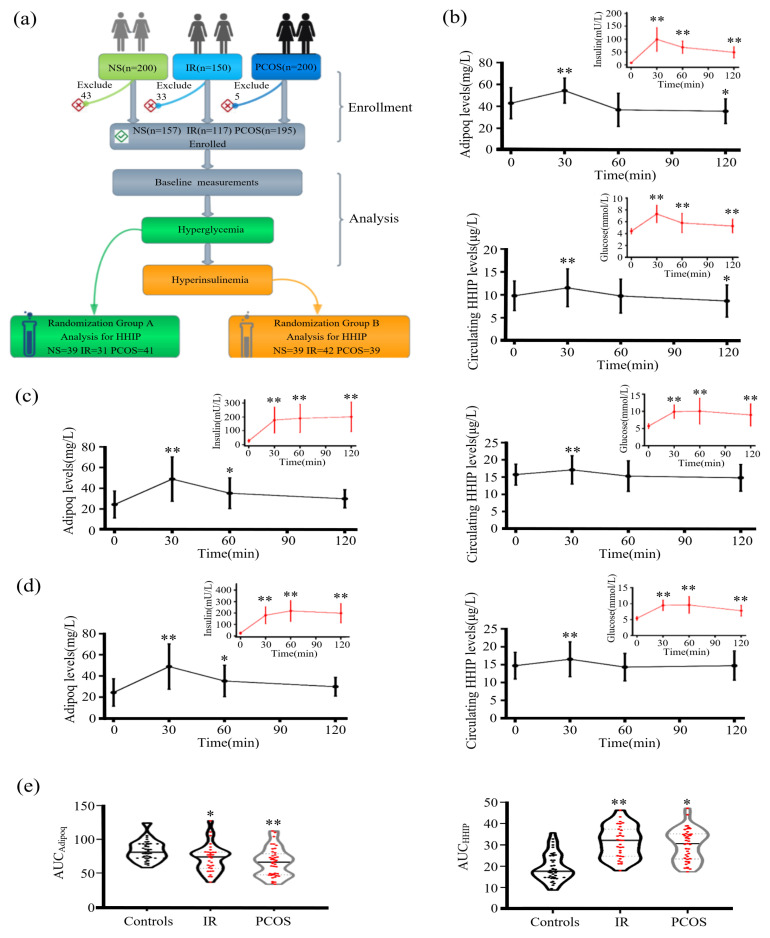
Serum HHIP levels in the study population during the OGTT. (**a**) Experimental design for the cross-sectional study. (**b**–**d**) Time course of serum HHIP and adipoq changes during the OGTT in control (**b**), IR (**c**) and PCOS (**d**) women. (**e**) AUC_HHIP_ and AUC_adipoq_ during the OGTT in the study population. Data are expressed as mean ± SD or median (Interquartile Range). * *p* < 0.05, ** *p* < 0.01 vs. baseline controls.

**Figure 4 jcm-12-00888-f004:**
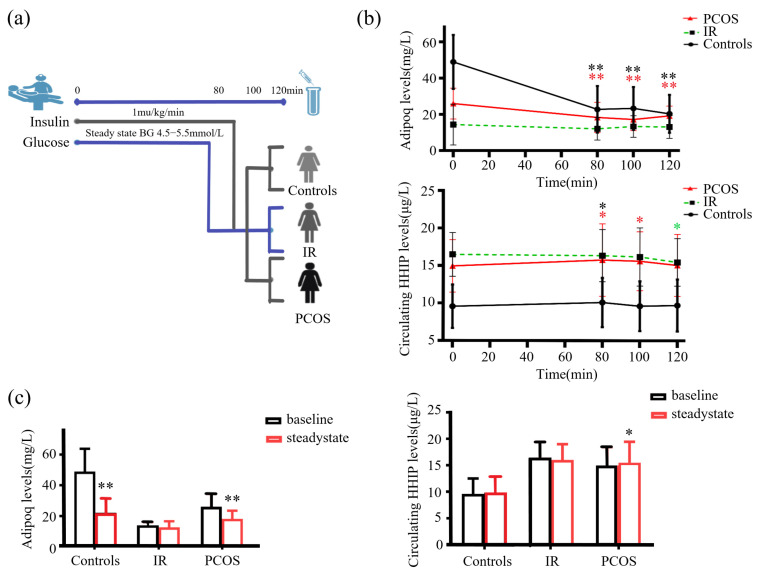
Serum HHIP levels in the study population during the EHC. (**a**) Experimental design for EHC. (**b**) Changes in Adipoq and HHIP levels during the EHC. (**c**) Cumulative Adipoq and HHIP levels during the EHC. Data are expressed as mean ± SD. * *p* < 0.05, ** *p* < 0.01 vs. baseline controls.

**Figure 5 jcm-12-00888-f005:**
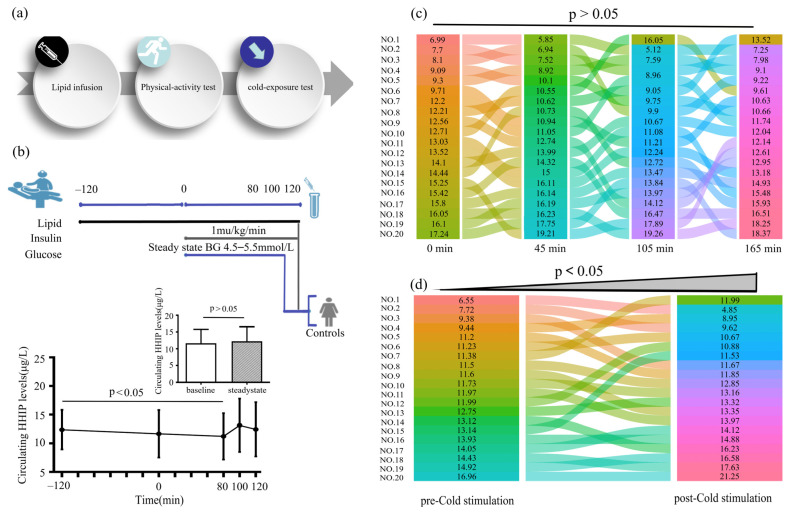
Circulating HHIP levels PCOS women or healthy controls during the intervention studies. (**a**) Intervention experimental procedure. (**b**) Experimental design for lipid infusion and EHC (upper panel), and changes in circulating HHIP levels during lipid infusion in healthy subjects (below panel). (**c**) Changes in circulating HHIP levels before and after exercise experiments in normal individuals. (**d**) Changes in circulating HHIP levels before and after cold-exposure experiments in normal individuals. Data are expressed as mean ± SD.

**Figure 6 jcm-12-00888-f006:**
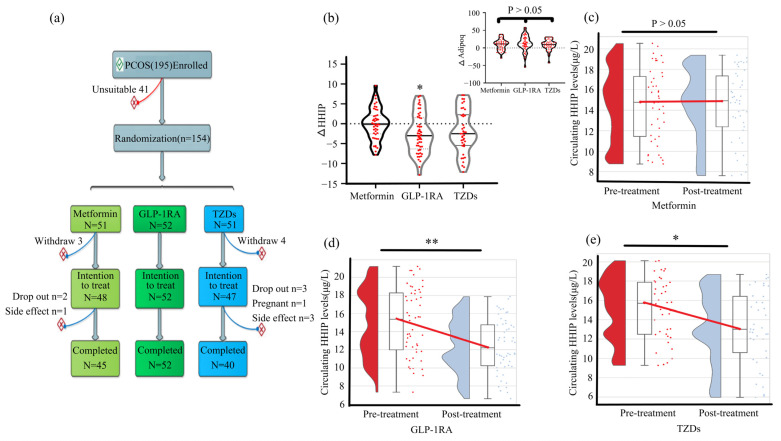
Effect of anti-diabetic drug therapy on circulating HHIP levels in women with PCOS. (**a**) Drug treatment protocol in women with PCOS. (**b**) Changes of circulating HHIP levels before and after treatment with anti-diabetes drugs. (**c**–**e**) Comparison of circulating HHIP levels pre- and post-metformin (**c**), GLP-1RA (**d**), and TZDs (**e**) treatment. Data are expressed as median (Interquartile Range). * *p* < 0.05, ** *p* < 0.01 vs. pre-treatment.

**Table 1 jcm-12-00888-t001:** Clinical, hormonal and metabolic parameters in the study population.

Parameter	Controls (*n* = 157)	IR (*n* = 117)	PCOS (*n* = 195)
Age (yr)	26 (24–28)	30 (27–33) **	27 (23–31) ^∆^
BMI (kg/m^2^)	20.1 (18.6–21.9)	25.5 (23.9–28.3) **	25.7 (22.8–28.6) **
WHR	0.78 (0.74–0.81)	0.86 (0.81–0.90) **	0.87 (0.82–0.91) **
SBP (mmHg)	106 (100–112)	116 (109–128) **	118 (108–121) **
DBP (mmHg)	71 (64–78)	75 (69–84) **	71 (67–78) ^▲^
TG (mmol/L)	0.98 (0.70–1.65)	0.99 (0.64–1.59)	1.53 (1.11–2.21) **^∆^
TC (mmol/L)	4.02 (3.41–4.50)	5.10 (4.52–5.50) **	4.64 (4.01–5.12) **^∆^
HDL-C (mmol/L)	1.29 (1.07–1.52)	1.18 (0.96–1.54)	1.17 (1.00–1.33) *
LDL-C (mmol/L)	2.29 (1.79–2.68)	3.20 (2.54–4.55) **	2.61 (2.10–3.19) **^∆^
FFA (μmol/L)	0.50 (0.36–0.69)	0.81 (0.52–1.11) **	0.52 (0.40–0.68) ^∆^
FBG (mmol/L)	4.52 (4.31–4.89)	5.43 (5.11–5.95) **	5.32 (4.97–5.71) **
2h-PBG (mmol/L)	5.05 (4.46–5.79)	8.09 (6.86–9.29) **	7.91 (6.65–9.03) **
FIns (mU/L)	7.01 (5.94–9.11)	18.33 (12.02–25.49) **	18.49 (13.12–29.57) **
2h-Ins (mU/L)	27.5 (17.6–52.5)	140.6 (98.8–279.8) **	145.4 (104.4–235.6) **
HbA1c (%)	5.1 (5.0–5.3)	5.4 (5.1–5.7) **	5.5 (5.2–5.7) **
AUC_g_	11.6 (10.3–13.3)	17.3 (15.0–19.7) **	16.8 (15.3–19.0) **
AUC_i_	98.8 (64.9–137.4)	254.6 (170.7–410.8) **	287.6(200.6–418.1) **^▲^
M-value (mg/kg/min)	9.54 (8.01–11.46)	4.16 (3.38–5.12) **	4.18 (3.31–5.49) **^▲^
HOMA-IR	1.39 (1.15–1.93)	4.49 (2.92–6.49) **	4.48 (3.02–7.01) **
VAI	1.41(0.92–2.36)	1.64 (0.86–2.82)	2.42 (1.68–3.71) **
BAI	27.3(25.0–28.8)	31.3 (29.3–33.7) **	30.7 (27.9–33.3) **
DHEA-S (μg/dL)	183.0 (146.4–221.5)	227.9 (174.6–299.4) **	237.2 (173.5–315.2) **
E2 (ng/L)	47.2 (27.3–67.6)	45.3 (33.2–55.1)	41.7 (31.8–56.6)
LH (IU/L)	4.55 (3.21–6.44)	4.00 (2.97–6.14)	7.58 (4.14–11.57) **^▲^
FSH (IU/L)	7.57 (6.55–9.29)	6.46 (5.29–7.95) **	6.18 (5.34–7.40) **
Prog (μg/mL)	2.18 (0.74–3.43)	1.30 (0.98–1.90)	1.68 (1.03–2.48)
SHBG (nmol/L)	56.8 (41.8–72.3)	35.0 (25.3–51.7) **	32.6 (23.5–46.0) **
TEST (nmol/L)	1.66 (1.19–2.11)	1.30 (1.06–1.85) *	2.08 (1.40–2.56) **^∆^
FAI	2.74 (1.78–4.46)	3.80 (2.36–6.23) **	6.22 (3.72–9.03) **^∆^
Adipoq (mg/L)	45.0 (38.3–55.7)	23.9 (9.0–32.2) **	25.8 (18.9–33.1) **^▲^
HHIP(μg/L)	9.24 (7.13–11.75)	15.50 (12.82–18.46) **	14.72 (11.81–17.77) **
HHIP ^§^	9.46 ± 0.26	15.57 ± 0.31 **	14.72 ± 0.23 **

PCOS, polycystic ovary syndrome; IR, Insulin resistance; BMI, body mass index; WHR, waist-to-hip ratio; SBP, systolic blood pressure; DBP, diastolic blood pressure; TG, triglyceride; TC, total cholesterol; HDL-C, high-density lipoprotein cholesterol; LDL-C, low-density lipoprotein cholesterol; FFA, free fatty acids; FBG, fasting blood glucose; 2h-PBG, 2h-post-glucose load blood glucose; FIns, fasting plasma insulin; 2h-Ins, 2h-plasma insulin after glucose overload; AUC_g_, the area under the curve for glucose; AUC_i_, the area under the curve for insulin; HOMA-IR, HOMA-insulin resistance index; VAI, visceral adiposity index; BAI, body adiposity index; DHEA-S: Dehydroepiandrosterone sulfate; E2, estradiol; LH, Luteinizing hormone; FSH, Follicle- stimulating hormone; Prog, Progesterone; SHBG, Sex-hormone binding globulin; TEST, testosterone; FAI, free androgen index = T(nmol/L)/SHBG (nmol/L) × 100. Adipoq, adiponectin. ^§^ Mean ± SE by general linear model with adjustment of age. Values are given as median (Interquartile Range) or mean ± SE. * *p* <0.05, ** *p* < 0.01 compared with controls; ^▲^
*p* <0.05, ^∆^
*p* < 0.01 compared with IR group.

**Table 2 jcm-12-00888-t002:** Association of circulating HHIP with IR and PCOS in fully adjusted models.

Model Adjust	IR			PCOS		
	OR	95% CI	*p*	OR	95% CI	*p*
Age	1.768	1.580–1.978	<0.001	1.652	1.494–1.828	<0.001
Age, SBP	1.749	1.556–1.967	<0.001	1.632	1.469–1.813	<0.001
Age, SBP, DBP	1.750	1.554–1.971	<0.001	1.633	1.467–1.817	<0.001
Age, SBP, DBP, BMI	1.727	1.517–1.966	<0.001	1.607	1.425–1.812	<0.001
Age, SBP, DBP, BMI, WHR	1.700	1.492–1.936	<0.001	1.586	1.405–1.791	<0.001
Age, SBP, DBP, BMI, WHR, TG	1.705	1.496–1.942	<0.001	1.598	1.414–1.806	<0.001
Age, SBP, DBP, BMI, WHR, TG, TC	1.708	1.493–1.954	<0.001	1.592	1.406–1.803	<0.001
Age, SBP, DBP, BMI, WHR, TG, TC, HDL	1.716	1.498–1.966	<0.001	1.603	1.414–1.818	<0.001
Age, SBP, DBP, BMI, WHR, TG, TC, HDL, LDL	1.731	1.507–1.989	<0.001	1.593	1.404–1.807	<0.001
Age, SBP, DBP, BMI, WHR, TG, TC, HDL, LDL, FFA	1.780	1.538–2.061	<0.001	1.620	1.419–1.849	<0.001

## Data Availability

Not applicable.

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
