# Peer review of "Circulating HHIP Levels in Women with Insulin Resistance and PCOS: Effects of Physical Activity, Cold Stimulation and Anti-Diabetic Drug Therapy"

_jcm, 2023, doi:10.3390/jcm12030888_

Round 1

Reviewer 1 Report

Dear Authors,

 The work presented to me for evaluation is very interesting and meets the thematic requirements of the journal. The work is prepared very carefully, but some technical errors need to be corrected. Please review the paper carefully and comply with the requirements of the journal, especially:

- please explain the purpose of using men in the experiment in section 2.7.

- KEYWORDS: it is recommended to add explanations of abbreviations

- some figures (Figure 2,3 and 4) are too small, so they are hard to read - please enlarge them

- no descriptions for Figure: 3f, 3g, 3h - under the figure

- title of figure 4 - figure 4j is missing or the description is incorrect

-no description for Figure 4e in text

- all abbreviations used in the tables should be explained, also in the case of supplementary material.

Author Response

Point 1: Please explain the purpose of using men in the experiment in section 2.7.

Response 1: To exclude the effect of gender on circulating HHIP levels, we also observed the effect of cold-induced adaptive thermogenesis on circulating HHIP in healthy men. This content has been further explained in the Results (See page 13).

Point 2: KEYWORDS: it is recommended to add explanations of abbreviations.

Response 2: As required, add explanations of abbreviations have been added to KEYWORDS (See page 1).

Point 3: Some figures (Figure 2, 3 and 4) are too small, so they are hard to read - please enlarge them.

Response 3: As required, these Figures have been enlarged, and Figure 3 is divided into Figure 3 and 4 (See page 12-13). Figure 4 is divided into Figure 5 and 6 (See page 14-15).

Point 4: no descriptions for Figure: 3f, 3g, 3h - under the figure.

Response 4: We are sorry for these omissions, and these descriptions have been added to Figure: 4a, 4b, 4c (See page 13).

Point 5: title of figure 4 - figure 4j is missing or the description is incorrect.

Response 5: We apologize for the mistake, and they have been corrected (See page 15).

Point 6: no description for Figure 4e in text.

Response 6: We apologize for this omission, and it has been descripted as Figure 5a (See page 15).

Point 7: All abbreviations used in the tables should be explained, also in the case of supplementary material.

Response 7: As required, all abbreviations have been explained (See all table).

Reviewer 2 Report

As PCOS affects up to 1 in 10 women there is a constant drive to better regulate treatment. Thus, this study is most welcome as it consolidates information on the best therapeutic approach. This study will, as the authors acknowledge, need further work and subsequent study designs to finer tune conclusions. This study has great merit and a lot of work has been imbued within.

Author Response

Point 1: As PCOS affects up to 1 in 10 women there is a constant drive to better regulate treatment. Thus, this study is most welcome as it consolidates information on the best therapeutic approach. This study will, as the authors acknowledge, need further work and subsequent study designs to finer tune conclusions. This study has great merit and a lot of work has been imbued within.

Response 1: We thank and appreciate the reviewer's comments.

Reviewer 3 Report

The authors investigated human hedgehog-interacting protein (HHIP) as a marker of metabolic abnormalities in PCOS women. Using several biometric and laboratory examinations as well as oral glucose test, euglycemic-hyperinsulinemic clamp experiments the metabolic status of the healthy controls and women with PCOS was assessed. Moreover some of the helthy controls had lipid infusion, exercise interventions and cold-exposure procedure to check how different stili affect HHIP.

The next step of the investigation examined how interventions with metformin, GLP-1RA and thiazolidinedione change metabolic status of PCOS women.

The study shows the possible potential of HHIP as a metabolic marker in policystic ovary syndrome.

The study was well designed and performed. In line 132 there is a mistake. It shoud be probably "backed to 270C again".

Author Response

Point 1: The study shows the possible potential of HHIP as a metabolic marker in policystic ovary syndrome. The study was well designed and performed. In line 132 there is a mistake. It shoud be probably "backed to 270C again".

Response 1: We appreciate and agree with the reviewer's comment. We apologize for this mistake in line 136, and it has been revised (See page 3).

Reviewer 4 Report

This study focuses on the complexity of the PCOs’ related endocrinological/metabolical abnormalities, first of all IR, in the search new biomarkers needed for accurate diagnosis and treatment. The authors developed the study on different platforms: starting with bioinformatics (to explore the association of PCOS with metabolic-related genes and signaling pathways), they carried on an observational cross-sectional study on more of 469 individuals to support the hypothesis of the involvement of serum HHIP and adiponectin level on the prevalence of serious PCOs or IR, hypothesis further confirmed by the following ROC curve analyses for the prediction of PCOS or IR. In addition, they also performed OGTT and EHC experiments on all the subjects enrolled, testing serum HHIP and adiponectin concentrations also.

In additional 20 healthy volunteers the authors observe the effect of increased FFA concentration on serum HHIP level, by lipid infusion and EHC, or physical activity and cold-exposure test. Moreover, 137 women with PCOS were treated with metformin, GLP-1RA, or TZDs for 24 weeks. Serum HHIP levels were higher in insulin resistance subjects and PCOS women. Circulating HHIP levels were significantly correlated with adiponectin levels, obesity, IR, and metabolic indicators.

Results showed that circulating HHIP levels were associated with glucose and lipid metabolism and insulin sensitivity, not affected by 45-min aerobic exercise but affected by cold exposure in healthy women. Treatments with GLP-1RA and TZDs in women with PCOS improved IR, increased serum adiponectin levels, and decreased HHIP levels. In this way they propose HHIP as a potential biomarker for the identification of high-risk candidates among women with IR and PCOS.

The authors developed this study in multiple floors, from bioinformatics analysis to observational cross-sectional study, with additional short-term intervention studies and follow-up studies. They succeeded to demonstrate the association of PCOS with metabolic-related genes and signaling pathways such as HHIP and DHEA-S, FAI, SHBG, and FSH, and, more interestingly, they showed that lipid infusion decreased serum HHIP levels, while cold exposure increased HHIP levels in healthy women. GLP-1RA and TZDs treatment reduced serum HHIP levels in PCOS women, while metformin treatment did not affect HHIP levels.

This is a well-thought -developed and presented study. The authors really did a great job both for the large cohort of individuals analyzed and for the analysis/treatments/experiments performed.

Good job!

Minor

Lane 86-88: The sentence: “The healthy women as control group participated in physical examinations, and the school and community volunteers” is not clear.

Author Response

Point 1: Lane 86-88: The sentence: “The healthy women as control group participated in physical examination, and the school and community volunteers” is not clear.

Response 1: We thank and appreciate the reviewer's comments. We apologize for this mistake, and it has been corrected (See page 2).